# Metabolism of Glycosphingolipids and Their Role in the Pathophysiology of Lysosomal Storage Disorders

**DOI:** 10.3390/ijms21186881

**Published:** 2020-09-19

**Authors:** Alex E. Ryckman, Inka Brockhausen, Jagdeep S. Walia

**Affiliations:** Department of Biomedical and Molecular Sciences, Queen’s University, Kingston, ON K7L 2V5, Canada; 14aer1@queensu.ca

**Keywords:** glycosphingolipids, biosynthesis, glycosyl hydrolases, lysosomal storage diseases, degradation of glycoconjugates, pathophysiology

## Abstract

Glycosphingolipids (GSLs) are a specialized class of membrane lipids composed of a ceramide backbone and a carbohydrate-rich head group. GSLs populate lipid rafts of the cell membrane of eukaryotic cells, and serve important cellular functions including control of cell–cell signaling, signal transduction and cell recognition. Of the hundreds of unique GSL structures, anionic gangliosides are the most heavily implicated in the pathogenesis of lysosomal storage diseases (LSDs) such as Tay-Sachs and Sandhoff disease. Each LSD is characterized by the accumulation of GSLs in the lysosomes of neurons, which negatively interact with other intracellular molecules to culminate in cell death. In this review, we summarize the biosynthesis and degradation pathways of GSLs, discuss how aberrant GSL metabolism contributes to key features of LSD pathophysiology, draw parallels between LSDs and neurodegenerative proteinopathies such as Alzheimer’s and Parkinson’s disease and lastly, discuss possible therapies for patients.

## 1. Introduction.

Glycolipids consist of at least one sugar residue covalently linked to a lipid moiety and can be categorized into two groups: (1) glyceroglycolipids, which are composed of a glycerol backbone, at least one fatty acid and a carbohydrate, and (2) glycosphingolipids (GSLs), composed of an N-acylated, unsaturated amino alcohol sphingosine (ceramide) and at least one carbohydrate residue. GSLs are amphipathic due to the presence of a variable hydrophilic carbohydrate-rich head group and a hydrophobic ceramide tail. A high degree of diversity exists in GSL structure and function. GSLs can be further classified by charge, where neutral and acidic GSLs possess uncharged and anionic carbohydrate constituents respectively. Glucuronoglycosphingolipids, sulfatoglycosphingolipids, phospho- and phosphono-glycosphingolipids and gangliosides constitute the anionic group of GSLs.

Glyceroglycolipids are infrequently found in humans, but are ubiquitous throughout plant cells. In contrast, GSLs are highly expressed throughout the central nervous system (CNS) of vertebrates [1,2]. In the brain and spinal cord of vertebrates, GSLs primarily populate lipid rafts within the plasma membrane along with cholesterol and proteins such as receptors, signaling molecules and glycosylphosphatidylinositol (GPI)-anchored proteins. The hydrophobic ceramide moiety anchors the GSL into the membrane, while the hydrophilic glycan extrudes into the extracellular space and can interact with other membrane-bound and extracellular carbohydrate-binding molecules. Overall, the biosynthesis, degradation and intracellular trafficking of GSLs are tightly regulated as these glycolipids serve several critical functions for the cell. Perturbation in GSL homeostasis results in serious pathological consequences and is the cause of various metabolic disorders, which will be discussed later in the current review.

## 2. Glycosphingolipids

### 2.1. Glycosphingolipid Structures

Among the simplest GSLs are Galβ-ceramide (GalCer) and Glcβ-ceramide (GlcCer), which are both highly abundant in the vertebrate brain (Figure 1A–E). Further processing of GalCer and GlcCer can involve the addition of neutral or acidic sugars. The addition of a Galβ1-4 residue forms lactosylceramide (LacCer)—the structural basis for Globo-, Lacto-, Neolacto- and Ganglio- series GSLs (Figure 1A–E). The 3-position of the Gal residue of GalCer may carry a sulfate ester in sulfatide [3]. Along with GalCer, sulfatides are also highly expressed in oligodendrocytes and myelin [4]. Studies in mice have illustrated that GalCer is essential for neural development, and maintenance of stability and the protective functions of myelin in both the central and peripheral nervous system. Additionally, attachment of sialic acid in an α2-3 linkage to GalCer forms GM4, a myelin-specific ganglioside and the foundation of the Gal a-series GSLs (Figure 1B). 

The Lacto-series GSLs are based on GlcNAcβ1-3Galβ1-4Glcβ-Cer, and Globo- series have Galα1-4 Gal linkages, e.g., in Galα1-4Galβ1-4Glcβ-Cer (Figure 1A,C). Ganglio- series GSLs are modified by GalNAc instead of GlcNAc and are based on GalNAcβ1-4 Galβ1-4Glcβ-Cer, which can be extensively sialylated. These sialylated GSLs (gangliosides) are common throughout the nervous system and are essential components of membrane rafts.

It is evident that a variety of modifications can be made to GSLs and one such example is the production of blood group antigens. The most well-known antigens are the ABO blood group antigens, composed of an H-antigen backbone (Type O), glycosylated with either GalNAcα1-3 (Type A), Galα1-3 (Type B) or both (Type AB). The little i and big I antigens, P^k^ antigen, Forssmann and Lewis antigens are found on GSLs, as well as on N- and O-glycans of glycoproteins, which coat the surface of red blood cells. Overall, many of these antigens serve as recognition motifs for the immune system during times of infection and cancer development.

The lipid moieties of GSLs can vary in length of fatty acid chains and degree of unsaturation, and in turn contribute to the diversity of cellular behaviors exhibited by each GSL. Together with at least six sugars (Glc, Gal, GlcNAc, GalNAc, sialic acid and Fuc), bonded in several types of linkages, with different branches, acidity and modifications by sulfate esters—hundreds of GSL structures exist.

### 2.2. Gangliosides

Gangliosides are found on the cell membrane of all vertebrate cells, including the heart, skeletal muscle, liver, intestines and recently in certain immune cells, but are most prevalent within neuronal cells [2,5,6]. Gangliosides constitute up to 6% of the lipid weight of the mammalian brain, and of which, GM1, GD1a, GD1b and GT1b gangliosides are the most common [7]. Accordingly, gangliosides are 5 times more concentrated in the grey matter nuclei than in the white matter tracts. 

As previously mentioned, gangliosides are negatively charged by means of a sialic acid constituent. The sialic acid residues of gangliosides are α2-3 linked to Gal or α2-6 linked to GalNAc, and extension of the sialic acid residues are through sialyl α2-8 linkages. The type of sialic acid most frequently found in vertebrates is N-acetylneuraminic acid (Neu5Ac), a relatively strong acid (pKa 2.6). Although Neu5Ac cannot be converted to its derivative N-glycolylneuraminic acid (Neu5Gc) in humans, Neu5Gc has still been found present in human tissue and plasma. It is thought that Neu5Gc may be derived from ingestion of animal products, which contain this sugar. Additionally, the presence of O-acetyl groups on sialic acid, or the absence of acyl groups as seen in lyso-GSLs, modify GSL solubility [8,9], affecting both degradation and function. According to common nomenclature rules, the number of sialic acid groups bound to the inner Gal molecule (Figure 1E) dictates the class of the ganglioside. The a-series gangliosides have one, b-series have two and c-series have three sialic acid residues [10,11].

During early development of the CNS, GM3 and GD3 are the predominant gangliosides, which are subsequently metabolized into complex gangliosides such as GM1, GD1a, GD1b and GT1b [11]. Interestingly, the ganglioside profile of a subject changes with age. In an adult spinal cord, GD1b and GD3 are among the most prevalent gangliosides throughout the anterior and posterior regions of both the cervical and lumbar spinal cord [12]. The total concentration of GM2 and GM3 ganglioside is less than 5%. In adult mice, GM1 is mostly found in white matter tracts and parallels the expression pattern of myelin-associated glycoprotein (MAG), a marker for myelin [13]. In comparison, GD1a and GT1b are more concentrated in the grey matter nuclei. Considering the neural localization of gangliosides, disruption of ganglioside metabolism can lead to a severe neurological phenotype, as exhibited in lysosomal storage disorders (LSDs) such as Tay–Sachs (TSD) and Sandhoff disease (SD).

The ganglioside patterns in mice mimic those in a human newborn with SD and TSD, and mouse models have greatly advanced our knowledge of ganglioside metabolism and therapy of LSDs [14]. However, in certain types of LSDs, the symptoms developing in mice were less severe than those in humans suggesting a greater complexity of the human syndrome [15]. For example, the SD and TSD disease models in mice do not completely recapitulate the human phenotype. In the TSD model, *Hexa*
^-/-^ mice do not suffer neurological or behavioral deficits, likely due to an alternative pathway capable of catabolizing GM2 ganglioside [16,17]. In contrast, the SD mouse model is quite severe and causes GM2 storage with marked neurodegeneration within the CNS. The SD model reflects more of a juvenile/adult-onset variant, as symptoms typically do not manifest until 16 weeks of age in mice—the equivalent of a young adult. Future research should look to disrupt both *Hexa/b* as well as *Neu3* genes to produce a more severe infantile model of GM2 gangliosidosis.

### 2.3. Isolation and Analysis of Glycosphingolipids

The methods for extracting GSLs from sample tissue ha have essentially remained the same since the establishment of the Folch protocol in the 1940s. Typically, chloroform/methanol/water mixtures are used to solubilize and extract GSLs from lipid membranes and vesicles. While the hydrophobic ceramide of GSLs is soluble in organic solvents, the solubility of an individual GSL is controlled by the number of its hydrophilic sugar residues. Further purification of specific GSLs and gangliosides can be performed with anionic exchange, reversed-phase liquid chromatography (HPLC) or high performance anion exchange chromatography (HPAEC) [18].

A number of tools can be used to analyze gangliosides and determine their structures. High performance thin layer chromatography with a hydrophobic solvent can identify GSL species by comparison to a standard GSL. Liquid chromatography–tandem mass spectrometry (LC–MS/MS) methods are an excellent tool to semiquantitatively determine the type of GSL present in a mixture, (e.g., in plasma and serum samples) [19,20,21]. Additionally, nuclear magnetic resonance (NMR) spectroscopy in deuterated hydrophobic solvents can identify individual sugars of a GSL, as well as their specific linkages. Lastly, glycosidases naturally cleave specific terminal sugar residues, and therefore are useful for identifying the sequences of glycan chains. Antibodies, lectins and other carbohydrate-binding proteins are also valuable tools to isolate and identify specific GSLs.

### 2.4. Functions of Glycosphingolipids

The biosynthetic enzymes that produce the lipid moieties of a GSL control both the structure and presentation of a GSL in the membrane, thereby regulating the type of intramembrane interactions it may engage in. In comparison, glycosyltransferases (GTs) control the structure of the hydrophilic glycan epitope, which extends into the extracellular environment and interacts with proteins such as lectins. Therefore, GTs govern cis or transcellular interactions, cell adhesion and signal transduction carried out by a GSL [22]. GSLs perform their functions remotely within lipid rafts on the plasma membrane and control the architecture of rafts by forming a number of complexes. Within the lipid raft, receptors, cell adhesion molecules and ligands exist in a dynamic fashion [23]. The balance of biosynthetic and degradation pathways changes upon ageing, affecting levels and distributions of GSLs and their role during development, differentiation and in cancer [1].

The smallest GSL and biosynthetic intermediate for most GSLs is GlcCer, a critical player in the development of the nervous system [24]. In a cystic fibrosis (CF) mouse model, GlcCer has been shown to reduce inflammation after inhalation of *Pseudomonas aeruginosa* [25], consequently preventing severe pneumonia. This parallels the finding that an inhibitor of glucosylceramidases (GBA1 and GBA2) also blocks inflammation in infected CF mice. The patterns of GSLs change in mouse brains during development, likely due to altered expression of anabolic and catabolic enzymes [26].

Using cultured bronchial cells from CF patients, inhibition of GlcCer degradation prevented the inflammatory response measured by IL-8 mRNA expression after infection with *Pseudomonas aeruginosa* [27]. Inhibition of GlcCer production also reduces the concentration and function of other related gangliosides. However, the individual ceramide unit of GlcCer activates apoptosis in salivary carcinoma cells [28] as well as tumor necrosis factor-α (TNFα)-stimulated cultured lung endothelial cells [29].

GT1b, GD1a, GM3 and GM1 have been shown to inhibit cell proliferation via inhibition of the phosphorylation of the epidermal growth factor receptor (EGFR) [30]. Specifically, the addition of GM1 to cultured human breast cancer cell lines decreased cell proliferation and EGFR activation [31]. It has been proposed that GM1 translocates the EGFRs from GSL-rich microdomains to small membrane invaginations known as caveolae, and therefore spatially separates the receptor from its ligand. Caveolae are abundant on most vertebrate cells, and are highly concentrated in adipocytes. In fact, the function of adipocyte insulin receptors hinges on an interaction with caveolin-1. In a state of insulin resistance stimulated by TNFα, it has been illustrated that GM3 causes dissociation of the insulin receptor from caveolae and inhibits its function [32,33].

Gangliosides are major contributors to neurodevelopment as they support formation and stabilization of functional synapses, creating the foundation for memory and learning [2,5,11]. It is evident that dietary ganglioside consumption during the early postnatal period increases cognition of infants by enhancing synaptic plasticity in the hippocampus and nigro-striatal pathway [34]. In neuroblastoma cells, the complex of GM1 with Gal-binding Galectin-1 stimulates a cellular shift in behavior from proliferation to differentiation [35]. It is now clear that GM1 increases activity of Ret tyrosine kinase, which in turn activates a cellular cascade resulting in neurotrophic action [36]. GM1 enhances the binding of the glial-derived neurotropic factor (GDNF) to its coreceptor GFRα1 in striatal dopaminergic neurons.

Interestingly, GSLs have been implicated in many cancers as high expression of GD2 and GD3 in neuroectoderm derived tumor cells appears to regulate cancer stem cell maintenance, invasion and metastasis [37] and thus are potential targets for anticancer treatments. Additionally, a family of soluble glycolipid transfer proteins (GLTP) allows intermembrane transfer of GSLs. The GSL binding site of GLTP consists of an α-helical, two-layer sandwich hydrophobic pocket that accommodates GSLs of different lengths and structures [38]. This binding helps to regulate GSL homeostasis and transport, and may be of therapeutic value in regulating GSL levels.

### 2.5. Microbial Interactions with Glycosphingolipids

Gangliosides, and specifically their Neu5Ac residues, serve as receptors for viruses including coronavirus, adenovirus, rotavirus, polyomavirus and Sendai virus. GD1a and GT1b act as receptors for a murine polyomavirus and control viral entry and infection of the cell [39]. In the absence of ganglioside receptors, viruses are still internalized into fibroblasts, possibly by a trafficking mechanism not involving gangliosides, but without subsequent infection.

Influenza viruses display glycoproteins hemagglutinin (HA) and neuraminidase (NA) on the viral surface that allows lateral movement of the virus [40]. Initially, the Neu5Ac residues of GSLs form attachment sites for Influenza A where binding of HA triggers clathrin-mediated endocytosis of the virus into the cell. Following infection, NA cleaves sialic acid from the membrane, which prevents other sialic acid-binding microbes from entering the same cell. Influenza C differs from other influenza viruses as it binds to 9-O-acetylated 5-N-acetyl neuraminic acid (Neu5,9Ac2). In addition, dendritic cells are antigen presenting cells that are essential for induction of immunity against viruses and other microbes. The lethal Ebola virus also appears to require gangliosides for entry into activated dendritic cells [41]. The sialic acid-binding lectin Siglec-1 recognizes gangliosides on the Ebola virus membrane and mediates viral uptake into the cytoplasm. This process can be blocked with an anti-Siglec-1 antibody. Siglec-1 does not distinguish between sialic acids in different linkages [42].

GSL-rich lipid rafts have been implicated in HIV-1 infections. The HIV-1 virus buds from microdomains on the plasma membrane, thus acquiring GSLs from ganglioside-rich lipid rafts. Host-derived GM3, but not GM1, on the HIV-1 envelope was shown to be involved in dendritic cell attachment of HIV-1 and infection [43,44]. The HIV-1 virus is also captured by dendritic cells through recognition of sialyl-lactose epitopes found on surface GSLs and glycoproteins [45].

Several bacteria are known to recognize the sialic acid residues of gangliosides, either via surface adhesins or hemagglutinins. For example, the asialo-GM1 ganglioside (GA1) serves as a receptor for *Pseudomonas aeruginosa* in the cornea of mice [46]. Anti-GM1 antibodies can be produced following infection with *Campylobacter jejuni,* which contain oligosaccharides that structurally mimic GM1. This process is the leading risk factor for the development of nerve dysfunction in Guillain–Barré syndrome. Interestingly, an unfortunate occurrence of this neurological disease was also seen in several people after vaccination with the swine flu vaccine contaminated with GM1 from *Campylobacter jejuni* [47,48].

Toxins produced by bacteria may also have an affinity for specific gangliosides. For example, *Vibrio cholerae* produces cholera toxin that specifically binds membrane-bound GM1 followed by cellular uptake into intestinal epithelial cells [49]. *Clostridium botulinum* neurotoxins bind to gangliosides as well as phospholipids [50]. The *C. botulinum* BoNT/C toxin binds to GD1b and GT1b as functional receptors, having di-sialyl glycan chains, whereas BoNT/D toxin induces toxicity in a ganglioside-independent manner. Shiga toxins or verotoxins are produced by a number of bacteria, including enterotoxigenic *Escherichia coli*. Shiga toxins STx1 and STx2 bind to Gb3 of the Globo series GSLs (Figure 1C), which contain the Galα1-4 linkage to LacCer [51]. Gb3 is abundant in the kidney and toxin binding can induce hemolytic uremic syndrome. Multimeric, water-soluble and adamantyl Gb3 analogs have shown promise in blocking the toxic effects of the Shiga toxin. It has also been suggested that the strong binding of Shiga toxin STx2 to Gb3 could be blocked by O-antigens containing a structural Gb3 analog [52].

Infections with the pathogenic fungus *Cryptococcus neoformans* in mice seem to depend on the presence of specific GlcCer structures. The fungus showed decreased virulence in mice after knock out of the gene encoding ceramide β-Glc-transferase. The loss of GlcCer was associated with altered cell membrane permeability that rendered the mutant fungi unable to grow within host macrophages [53].

## 3. Biosynthesis of Glycosphingolipids

The initial process of GSL biosynthesis at the cytoplasmic face of the endoplasmic reticulum (ER) membrane involves a condensation reaction of Ser and Palmitoyl-CoA yielding 3-keto-dihydrosphingosine, which is subsequently N-acylated to form Cer. The sphingosine moiety of Cer is usually hydroxylated and unsaturated in humans. Cer likely flips between the two sides of the ER membrane, possibly assisted by flippase proteins. Cer is then distributed to Golgi membranes, either through vesicular transport or assisted by the cytosolic protein GLTP that mediates non-vesicular intramembrane trafficking of GSLs [54,55]. Further extension of Cer takes place in the Golgi apparatus by membrane-bound GTs and sulfotransferases [56]. All of these transferases are Golgi resident type II membrane proteins and have at least one N-glycosylation site, with a short cytoplasmic amino terminus, a hydrophobic transmembrane domain near the amino terminus and a catalytic domain localized to the Golgi lumen [56,57]. The enzymes catalyzing various steps in the biosynthesis are found in various Golgi compartments including the trans-Golgi network. None of the human GTs involved in GSL biosynthesis (Table 1) have been crystallized as of yet. Therefore, GT protein structures and mechanisms are based on protein modeling predictions.

GTs have been classified by the carbohydrate active enzymes (CAZy) data bank based on their sequences, predicted folds and catalytic mechanisms. The GTs that assemble GSLs utilize nucleotide sugars as donor substrates for the transfer of sugar residues, and have at least one predicted nucleotide binding domain. Most of these GTs (Table 1) are inverting enzymes that invert the anomeric configuration of the nucleotide donor substrate in the reaction product. These donor substrates are synthesized in the cytoplasm and include UDP-Glc, UDP-Gal, UDP-GalNAc, UDP-GlcNAc, GDP-Fuc and the donor substrate for sulfation, 3’phospho-adenosine 5’phosphosulfate (PAPS). Specialized transporters carry these donor substrates into the lumen of the Golgi where they are used by the GTs at the luminal face of the Golgi membrane [58]. These transporters are also antiporters and transport the nucleotide reaction products back into the cytoplasm [59]. The transport of these donor substrates is essential for glycosylation reactions. A lack of the GDP-Fuc transporter results in leukocyte adhesion deficiency II-CDG with Fuc-deficient glycolipids and glycoproteins and dysfunctional adhesion of immune cells [60]. CMP-sialic acid required for the biosynthesis of gangliosides is synthesized in the nucleus and diffuses into the cytoplasm. The CMP-sialic acid synthases have peptide sequences required for transport of the enzyme into the nucleus [61]. A CMP-sialic acid transporter, CST, then transports CMP-sialic acid into the lumen of the Golgi. 

### 3.1. Biosynthesis of Simple Glycosphingolipids

The ganglioside precursor GlcCer is synthesized at the cytoplasmic face of the Golgi by GlcCer synthase (UGCG). Sphingomyelin and GlcCer synthase are colocalized in the medial/trans Golgi, which controls GlcCer synthesis. Complex formation between transferase proteins appears to be a general mechanism that enhances the biosynthesis of glycans [62,63]. GlcCer synthase is a member of the CAZy GT21 family, and is considered an inverting GT-A folded, transmembrane Golgi membrane enzyme. Its multiple hydrophobic sequences explain its specificity for Cer-containing substrates. A number of potent inhibitors have been developed for this important enzyme that could alleviate the accumulation of GSLs in disease, including D,L-threo-1-phenyl-2-hexadecanoylamino-3-pyrrolidino-1-propanol-HCl (PPPP) [64]. The Golgi lipid transport protein FAPP2 recognizes GlcCer by its glycolipid-transfer-protein homology domain and transports it across the membrane [65]. Thus, FAPP2 has a pivotal role in the synthesis of complex GSLs.

Ceramide βGal-transferase (CGT and UGT8) is a transmembrane protein of the ER membrane, which extends its catalytic domain into the ER lumen to synthesize Galβ-Cer (cerebroside) [66]. CGT has been classified as a GT1, having a GT-B fold. The large CGT protein has a signal peptide at the N-terminus and a hydrophobic domain at the C-terminus reminiscent of a type I membrane protein. Thus, GlcCer and GalCer synthases have different folds and only share 10.8% sequence identity. Although both synthases recognize ceramide and UDP-hexose, and invert the stereochemistry of the donor substrate, they differ in their mechanism of action. GlcCer synthase, but not Cer βGal-transferase, has a DxD motif common in inverting GTs [57].

Both, Cer and GalCer are transported from the ER through vesicular transport and fuse with the Golgi membrane for further processing. GalCer can be converted to GM4 by the transfer of sialic acid in an α2-3 linkage to Gal from the CMP-β-sialic acid donor catalyzed by α3-sialyltransferase ST3Gal V. This enzyme is the essential initial enzyme in ganglioside biosynthetic pathways.

Sulfatide is a major GSL in the brain and is enriched in myelin as well as in spermatozoa. In the synthesis of sulfatide, sulfate is transferred from PAPS to the 3-hydroxyl group of Galβ-Cer by the Golgi enzyme Cer sulfotransferase (CST) [4]. CST can also sulfate the Galβ residue of LacCer and other substrates linked to a lipid-like group. Although CST has low activity towards LacCer and compounds having terminal Glcβ residues, it cannot act on substrates having terminal Galα substrates. Other rare structures based on GalCer include Galα1-3Galβ-Cer and GlcNAc-Galβ-Cer that may be synthesized by α3-Gal- and β3-GlcNAc-transferases of the Globo and Lacto series pathways (Figure 1A,C).

### 3.2. Glycosyltransferases that Extend Glycan Chains of Glycosphingolipids

Most of the GTs involved in GSL synthesis listed in Table 1 are inverting GTs that utilize a direct displacement S_N_2-like mechanism to transfer a sugar residue from a donor substrate to an acceptor substrate. The predicted mechanism involves a catalytic base that deprotonates the nucleophilic substituent of the acceptor to be glycosylated, facilitating the displacement of the phosphate leaving group in the donor substrate via a single oxocarbenium ion-like transition state. This results in the opposite anomeric glycosidic linkage in the reaction product, compared to that of the donor substrate. The catalytic base is commonly an Asp residue in the DxD sequon, assisted by bivalent transition metal ions such as Mn^2+^ or Mg^2+^.

The inverting β-Glc-, β-Gal-, β-GalNAc- and β-GlcNAc-transferases transfer sugars from the respective UDP-α-sugars to form β-linkages in the reaction products. Fuc-transferases use GDP-β-Fuc to synthesize Fucα-linkages and sialyltransferases use CMP-β-sialic acid to synthesize sialyl α-linkages in an ordered, sequential, stereospecific mechanism.

As seen in Figure 1A, β1,4-galactosyltransferase 6 (B4GALT6) synthesizes the Galβ1-4 linkage of LacCer on the luminal face of the Golgi membrane [67]. In mice, B4galt5 also synthesizes LacCer and the enzyme was found to be essential for development [68]. B4GALT5 and 6 are inverting GTs of the GT7 family with a predicted GT-A fold. LacCer is then converted further to GSLs of either the Lacto, Globo or Ganglio series [5] (Figure 1). The β1,3GlcNAc-transferase B3GNT5 (GT31) initiates production of the Lacto series that can carry multiple glycan epitopes such as blood group antigens [69]. The enzyme may also synthesize GlcNAc-GalCer. The addition of a Galβ1-3 residue to GlcNAc of Lacto series GSLs is carried out by B3GALT5 (Figure 1A). The GlcNAc residue of Neo-lacto series GSL is substituted by a Gal residue in β1-4 linkage. In the synthesis of globosides, Gb3 is synthesized from LacCer by α1,4-Gal-transferase A4GALT, followed by a transfer of GalNAc by β1,3-GalNAc-transferase B3GALNT1 to form Gb4 (Figure 1C).

The α3-sialyltransferases ST3Gal II and ST3Gal III transfer sialic acid to the terminal Gal residues in an α2-3 linkage. Further extension of the sialyl residues is catalyzed by α8-sialyltransferases ST8Sia I and ST8Sia V to form the complex, anionic ganglioside series carrying two or more sialic acid residues.

LacCer and the mono-, di- and tri-sialylated forms GM3, GD3 and GT3 are all substrates for N-acetylgalactosaminyltransferase B4GALNT1. This enzyme adds GalNAc in a β1-4 linkage to the Gal moiety, yielding GA2, GM2, GD2 or GT2, although the kinetics for the substrates differ. B4GALNT1 is a typical type II membrane protein. Subcellular localization studies using Chinese hamster ovary K1 cells showed that the enzyme localizes to the trans-Golgi network membranes [70]. The GalNAc residues can be further extended by the addition of Gal in a β1-3 linkage catalyzed by β1,3Gal-transferase 4 (B3GALT4) of the GT31 family, which yields GM1, GD1b and GT1c (Figure 1E).

Another sialyltransferase (ST6GALNAC V) acts exclusively on GalNAc residues and transfers sialic acid in an α2-6 linkage (present in α-series gangliosides) and converts GM1b to GD1α (Figure 1D). All human sialyltransferases belong to the same GT29 family and possess sialyl-motifs [71]. A number of GTs that are primarily involved in the extension of N- and O-glycan chains of glycoproteins can also utilize GSLs as substrates (e.g., ST3Gal I, IV and VI). 

The enzymes that synthesize blood group AB antigens are retaining GTs that preserve the stereochemistry of the donor substrates in the final product. However, α1,4-Gal-transferase (A4GALT), a GT32 family enzyme, that synthesizes the neutral globosides (Table 1) is the only retaining GT involved in GSL extension reactions. Several mechanisms have been proposed for retaining GTs, including a double-displacement mechanism involving the formation of a covalently bound glycosyl-enzyme intermediate. 

### 3.3. Genetic Defects of Glycosphingolipid Biosynthesis

While glycosidase gene defects are well studied and relatively common, human diseases of ganglioside biosynthesis are extremely rare and are classified as congenital disorders of glycosylation (CDG) [72]. Mutations of three genes encoding GTs are known to cause defects in GSL biosynthesis (Table 2). A deficiency of α2,3-sialyltransferase V (ST3Gal V-CDG) blocks the sialylation of LacCer to form GM3 and results in GM3 sphingolipodystrophy or Amish infantile epilepsy. The first case was reported in 1974, when an infant presented with severe motor impairment and persistent seizures culminating in death by 3.5 months of age [73]. A variant of GM3 synthase was also found in a patient that suffered from severe auditory, visual, motor and cognitive impairments, as well as respiratory chain dysfunction [74]. Due to the absence of ST3Gal V activity, a post-mortem examination of the patient’s brain revealed a complete absence of complex gangliosides. The symptoms of GM3 deficiency developing in mice were less severe than those in humans suggesting a greater complexity of the human syndrome [73].

In a patient with West syndrome (ST3Gal III-CDG), an illness characterized by epileptic spasms (ranging in severity), abnormal brain waves and intellectual disability [75], the α2,3-sialyltransferase III was found to be deficient, thus decreasing α2-3sialylation of Gal linked to GalNAc in GD1a, GT1b and GQ1c, which causes an accumulation of GM1. Missense variants of ST3Gal III have also been associated with epileptic encephalopathy and impaired neuromotor development in an infant [74]. In cortical neurons derived from induced pluripotent stem cells isolated from patients, a missense variant of ST3Gal III was associated with altered lectin binding patterns and enhanced adherence to poly-L-ornithine/laminin-coated surfaces [76]. The enzyme was found to be void of activity while other variants had very low activity [77].

Another defect was reported in the *B4GALNT1* gene, encoding B4GALNT1 which normally adds GalNAc to the Gal residue of LacCer, GM3 or other GSLs. A deficiency of β1,4-GalNAc-transferase (B4GALNT1-CDG) causes accumulation of GM3, GD3 and GT3 and a deficiency of GM2, GM1, GD2 and GSLs that carry the GalNAc residue (Figure 1E, Table 1). A frameshift insertion mutation was found to be the cause of a Kuwaiti family’s B4GALNT1 deficiency, which presented a phenotype of progressive spastic paraparesis and mild peripheral neuropathy [78,79]. The clinical features vary, depending on mutation severity and residual enzyme activity. Ultimately, it is clear that maintenance of GSL biosynthesis homeostasis is critical to human health. Mice having *B4galnt1* gene defects (*B4galnt1(+/-)* mice) show a partial deficiency of GM1 and are considered a model for Parkinson’s disease (PD). The neurological symptoms are comparable to those in human patients with *B4GALNT1* mutations [80]. The protein aggregation seen in these mice was successfully reduced by treatment with synthetic GM1 ganglioside, LIGA20, capable of crossing the blood–brain barrier (BBB) [81,82]. In human patients with complex hereditary spastic paraplegia, several mutations in the *B4GALNT1* gene were identified. The mutations involved a shift in GSL patterns associated with early-onset spastic paraplegia, intellectual disability, ataxia, cortical atrophy and peripheral neuropathy [83]. The efficacy of treatment with LIGA20 and GM1 remains to be confirmed in human patients.

## 4. Degradation of Glycosphingolipids

Biosynthesis, degradation and recycling of GSLs is are in constant dynamic equilibrium as these processes are required to maintain physiological membrane concentrations of GSLs. Membrane-bound GSLs are continually endocytosed and transported within the luminal face of endosomal vesicles where they are presented to hydrolases for sequential degradation. This catabolic process produces GSL constituents including monosaccharides and the components of ceramide, which are then recycled and reinserted into the bilayer membrane of lysosomes in a dynamic exchange.

The lysosomal hydrolases are targeted to lysosomes by means of their mannose-6-phosphate (Man-6-P) groups at the termini of N-glycans, which is then recognized by Man-6-P receptors. If properly folded, lysosomal enzymes move from the ER to the Golgi where GlcNAc-6-phosphate is transferred to terminal Man residues of N-glycans by lysosomal enzyme GlcNAc-6-phosphate-transferase. This is followed by cleavage of GlcNAc by GlcNAc-1-phosphodiester α-N-acetylglucosaminidase, which uncovers the Man-6-P marker. The Man-6-P receptor-lysosomal enzyme complex is then translocated to acidic endosomes followed by release of the hydrolase from the receptor and targeting to the lysosome. However, other minor pathways for lysosomal targeting also exist [84]. A number of enzymes are associated with the plasma membrane including β-hexosaminidase, β-GalNAc-transferase, β-galactosidase, β-glucosidase and ceramidase [85]. Sialidase I (NEU1) and β-galactosidase (BGAL) are found at the plasma membrane and form a complex with serine carboxypeptidase cathepsin A (CTSA and PPCA). CTSA has a chaperone-like function in complexing sialidase I and β-galactosidase while transporting them to the lysosomes where they can efficiently degrade GM1. Upon entry into the lysosome, CTSA cleaves the carboxy-terminal peptide from β-galactosidase, which mediates the oligomerization and activation of sialidase I and forms a stable, active three-enzyme complex [86]. 

The degradation of GSLs involves the sequential removal of sugar residues from the non-reducing end by glycosyl hydrolases (GHs; Table 3). In the lysosome, GHs are optimally active at an acidic pH of approximately 4-5. Since the lipid moieties of GSLs are buried in the membrane, small sphingolipid activator proteins, named saposins (SAPs), form complexes with the glycolipid substrates (e.g., GM2 or GA2) and render the GSL more accessible to GHs [87]. A precursor protein called prosaposin is processed into SapA, B, C and D variants that differ in affinity for specific GSLs. Thus, deficiency in the saposin precursor protein will affect a number of different enzymes and GSLs.

The GH families (CAZy database) are classified by sequence similarity, predicted structural folds and function (Table 3). Most GHs are retaining enzymes and may act through a double-displacement mechanism. A number of crystal structures aid in the classification of families and of clans that share a fold and catalytic mechanism. Members of the large clan GH-A are characterized by a (β/α)_8_ barrel fold, a retaining mechanism and two conserved Glu residues present in equivalent positions within the barrel, serving as an acid/base catalyst and a nucleophile. Similar to GTs, GHs yield either an inverted or retained glycosidic linkage in the reaction product. 

Sialidases (Neuraminidases NEU1-4) are CAZy GH33 family members and have a 6-fold β-propeller conformation. They cleave terminal sialic acid residues without the assistance of saposins. Modification of sialic acid by O-acetylation reduces the rate of hydrolysis by sialidases.

N-acetylhexosaminidases HexA and HexB are classified into the CAZy GH20 family of retaining hydrolases and a GH-K glycosyl hydrolase clan with a (β/α)_8_ barrel fold. The catalytic site uses the carbonyl oxygen of GalNAc as the nucleophilic base, with Glu as the proton donor. These enzymes interact with GM2-activator protein (GM2-AP) to facilitate the degradation of GM2. GM2-AP has an N-terminal and a central hydrophobic domain and binds to the anionic lysosomal membrane surface through multiple Lys residues, thus perturbing membrane structure and lifting GM2 out of the membrane for presentation to the hexosaminidase enzymes.

The β-galactosidase (BGAL, GLB1) is well studied [88] and cleaves Gal-residues from a variety of glycoconjugates including GM1. This retaining GH35, clan GH-A hydrolase has a (β/α)_8_ fold. Both, α-galactosidase and α-N-acetylgalactosaminidase belong to the GH27 family, clan GH-D and cleave Galα- and GalNAcα-linkages, respectively. GH27 family members have conserved catalytic Asp residues [89,90].

Glucocerebrosidase (glucoceramidase I, GCase I, GBA1, GH30 family clan GH-A) is an intracellular and membrane-bound hydrolase [85], which normally catabolizes GlcCer to Glc and Cer, and Glc-sphingosine to Glc and sphingosine, using Asp as the catalytic base. The GlcCer substrate presentation to GCase is facilitated by SapC. Following translation, GCase binds to its trafficking receptor, lysosomal integral membrane protein-2, causing targeting of GCase to the lysosome. Recently, it has been found that a transglucosylation reaction occurs for the lysosomal acid GCase to synthesize Glc-cholesterol and Gal-cholesterol [91]. The five N-glycans of GCase control its structure [92]. Glucocerebrosidase II (GCase II, *GBA2*) is a non-lysosomal enzyme found in the ER, Golgi and plasma membrane. This soluble enzyme is associated with cellular membranes, and also cleaves Glc from GlcCer, as well as Glc linked to bile acids. Mutations of the *GBA2* gene were identified in several patients with hereditary spastic paraplegia [93].

Moreover, SapB activates arylsulfatase, which hydrolyzes the sulfate ester of sulfatide. The Gal-cerebrosidase can then hydrolyze GalCer, assisted by SapA. This enzyme is a GH59 family member, clan GH-A, that is different from GCase I.

After degradation of the glycan chains, acid ceramidase (*ASAH1*) hydrolyzes ceramide into sphingosine and fatty acid [11]. Acid ceramidase binds SapC and SapD. The tetrameric structure of acid ceramidase shows a hydrophobic channel leading to the catalytic site [94,95].

## 5. Lysosomal Storage Disorders

Glycosphingolipidoses are LSDs characterized by the accumulation of GSLs, which exceed the normal storage capacity of the cell. This leads to alterations in the membrane and formation of abnormal lipid vesicles in a number of organs. Gangliosides mainly accumulate in the nervous system, due to their neuronal localization, leading to neuronal dysfunction. Pathogenic variants of the GH genes may cause enzyme misfolding or inactivation, resulting in low enzyme activity or change in substrate specificity [96,97]. Although the lack of catabolic activity partly affects glycoproteins, GSLs accumulate due to their hydrophobic nature, and are responsible for several pathologies (Table 4).

### 5.1. Gaucher and Krabbe Disease

Gaucher disease (GD) is the most common LSD with an incidence of 1 in 40,000 in the general population, although it is especially prevalent in the Ashkenazi Jewish population, with a prevalence of 1 in 1000 [98,99]. GD is an inherited autosomal recessive disorder of the gene encoding β-glucocerebrosidase (GCase I, GBA1) [100]. More than 350 mutations are associated with GD, where storage of GlcCer occurs mostly in the lysosomes of cells, mainly of the macrophage lineage, causing a wide range of manifestations. There are several different types of GD with toxic accumulation of GlcCer and other GSLs. The symptoms vary and can involve the brain, liver and spleen, bone marrow and other organs. Models to study GD and investigate treatment options have been developed, for example neurons from mice that lack the *GBA1* gene [101]. Substrate-reduction and enzyme replacement therapy have been successful in treating GD [102].

Similar to GD, Krabbe disease is an autosomal recessive disorder with mutation in the Gal-cerebrosidase gene (*GALC*)—in fact, 130 mutations have been identified. The most frequent form of Krabbe disease (KD) occurs early-onset in infants. Normally, Gal-cerebrosidase degrades GalCer, an essential constituent of myelin, as well as psychosine, a toxic byproduct of GalCer production. The resulting accumulation of GalCer and psychosine in the nervous system of patients causes progressive loss of myelin, resulting in symptoms related to the loss of neural function. Interestingly, psychosine accumulation is neurotoxic and has been reported to facilitate fibrillation of α-synuclein (SNCA) [99,100,103]. There is still no effective treatment for Krabbe disease, but stem cell transplant has been a modality of treatment in some states of the USA [104].

### 5.2. GM1-Gangliosidosis

GM1 gangliosidosis is an autosomal recessive disorder resulting in a deficiency of β-galactosidase due to the mutation of the *GLB1* gene [99,105,106]. Deficiency in BGAL activity affects the degradation of GSLs and glycoconjugates that have terminal β-Gal residues such as GM1, GA1 and mucopolysaccharides. GM1 gangliosidosis patients exhibit accumulation of GM1 in the grey matter up to 4-fold of GM1 in healthy individuals. Accordingly, neurodegeneration of the CNS represents a hallmark trait of pathology. In a mouse model of GM1 gangliosidosis it has been shown that accumulation of gangliosides, GA1 and GM1, affect the integrity of the mitochondrial membrane and may promote neurodegeneration [107]. GM1 also accumulates in the viscera, producing complications such as hepatomegaly, cardiomegaly, skeletal deformities, joint stiffness and muscle weakness. Since BGAL forms a complex with NEU1 and cathepsin A, the complex is compromised in the absence of BGAL. This can cause a secondary NEU1 deficiency that affects the degradation of gangliosides [106].

There are three presentations of GM1 gangliosidosis, which are classified according to the onset of disease. Type 1, also known as classic infantile GM1 gangliosidosis, is the most severe and arises around 6 months of age, where children begin to regress through developmental milestones only to rapidly deteriorate shortly thereafter. Affected children typically succumb to their illness by 2 years of age. Type 2 is milder, with onset between the ages of 1–5 years. These juvenile cases usually progress at a slower rate and present with identical symptoms along with ataxia, dementia and/or speech difficulties. The life expectancy of affected individuals is shortened, and patients die in late childhood or early adulthood. The least severe and slowest progressing variant is type 3 as symptoms usually develop later in adulthood. Enzyme replacement therapy has shown promise in patients with GM1 gangliosidosis [108]. Recently, an adeno-associated virus serotype 9 (AAV9) gene therapy has been developed for GM1 gangliosidosis [109] and clinical trials for this treatment began in 2019 (GM1 Gangliosidosis Gene Therapy Trial (AAV9-GLB1), Clinical trial No. NCT03952637. We are expecting more of these clinical trials. 

### 5.3. GM2-Gangliosidosis: Tay–Sachs and Sandhoff Diseases

GM2 gangliosidosis is a family of LSDs, which manifest due to deficient levels of β-hexosaminidase A resulting in failure to cleave the GalNAcβ residue from GM2, GD2, GT2 and Gb4 (Figure 1C,E) and the GlcNAcβ- residues from other GSLs. The variants of the GM2 gangliosidosis LSD family include TSD variants, SD and deficiency of the GM2 activator protein in AB variants [110], each characterized by mutations to one of the genes involved in GM2 catabolism. The ratio of accumulating gangliosides differs among patients, however, GM2 is the primary storage substrate found in neuronal lysosomes [111]. The degradation of GM2 and other GSLs depends on the activities of hydrolases. However, the lipid composition of membranes also controls the hydrolase activities as well as binding of activator proteins. Thus, altered membrane composition in LSDs affect the catabolism of GM2 [112]. Similar to GM1 gangliosidosis, affected individuals typically experience rapid neurological decline due to widespread neurodegeneration throughout the CNS.

There are three isoforms of hexosaminidase: HEXA occurs as an α,β heterodimer while HEXB is a β,β homodimer, and HEXS is an α,α homodimer. Both, the α- and β-subunits have an active site with slightly different substrate specificity; the α-subunit active site is coated with cationic residues enabling the accommodation of a negative charge such as that seen in GM2 [113]. Normally, the hexosaminidase α- and β-subunits associate in the ER and then are targeted to the lysosome for proper ganglioside degradation [114]. The GM2 activator protein (GM2AP) interacts with both the glycan and the lipid moiety of GM2 as well as the HEXA α-subunit, facilitating hydrolysis of GM2 by HEXA but not by HEXB [10,56,99]. Deficiency of either HEXA or GM2AP causes toxic GM2 accumulation in the CNS and plasma. It appears that residual HEXA activity of 5-10% is compatible with a lack of GM2 accumulation and a disease-free life.

TSD is the most common variant of GM2 gangliosidosis, with more than 100 mutations identified in the *HEXA* gene. Due to mutation of the α-subunit, HEXA and HEXS activities are reduced. Therefore GSLs having terminal GalNAc or GlcNAc residues accumulate, with mainly GM2, but also GA1, lyso-GM2 and other minor GSLs, promoting a lethal cascade of cellular events [112]. Diffuse neurodegeneration throughout the CNS is the result. The symptoms mirror GM1 gangliosidosis but do not involve visceral pathology. Affected individuals experience seizures, motor impairment, hearing and vision loss and eventually respiratory failure. TSD is characterized according to disease onset: classic infantile, juvenile and adult-onset. The most severe phenotype of TSD is infantile-onset as children begin to developmentally regress around 6 months of age resulting in death by age 4. The B1 variant of TSD shows deficiency in the association of the α and β subunits. The HEXA enzyme subunits have altered specificity where they are defective in GM2 degradation but still act on neutral GSLs, such as GA2 [110].

SD manifests as a result of mutation to *HEXB,* which encodes the β-subunit of HEXA and HEXB [115]. The β-subunit possesses stabilizing properties necessary for normal HEXA activity. Consequently, GM2, GA2 and lyso-GM2 gangliosides accumulate in the lysosomes of neurons that have deficient levels of or lack HEXA(α,β) or HEXB (β,β). SD presents with identical symptoms as TSD, with the addition of visceral manifestations. SD can also be subdivided into three classes based on disease onset with infantile cases presenting as the most severe and rapidly progressing condition.

Mice deficient in *hexa* or *hexb* genes were created as models for TSD and SD, respectively, with GSL accumulation and pathology largely similar to those observed in the human diseases. However, TSD mice lacking the α-subunit of HEXA and HEXB differ in pathological manifestations that did not involve the nervous system [116]. Interestingly, SD mice (*hexb* knockout) exhibit PD-like symptoms, with extensive SNCA aggregation in the brain and spinal cord, which increases with age. 

The third and most rare form of GM2 gangliosidosis is the AB variant, caused by pathogenic variants of the *GM2A* gene encoding GM2AP. GM2AP is a specific, required cofactor for HEXA [10] and is necessary for degradation of GM2. AB variant patients present symptoms similar to those of TSD and SD. There is no proven treatment for TSD/SD and patients are provided with supportive care only.

### 5.4. Other Lysosomal Storage Diseases in Brief

Table 4 lists additional gangliosidoses and sphingolipidoses, which are LSDs characterized by the accumulation of storage material within the lysosome, as well as their Mendelian inheritance in Man (MIM) numbers. In Fabry disease, the α-galactosidase GLA is deficient, leading to the accumulation of Galα-containing globosides, and affecting blood group B determinants of GSLs and glycoproteins. Many mutations have been found in GLA genes. The cell types involved depend on the secretor status (*Se* gene) that determines the occurrence of blood group B and AB determinants in epithelial glycoproteins throughout the body [117,118]. The GLA defect results in pain, kidney disease and heart defects and may include neuropathy and a number of other manifestations. 

Mutations of the α-N-acetylgalactosaminidase *NAGA* gene are the cause of the autosomal recessive Schindler (Kanzaki) disease. NAGA cleaves the GalNAcα-linkage of the blood group A determinants in GSLs and glycoproteins in individuals with blood group A and AB. The blood group A determinants accumulate throughout the body in individuals carrying the *Se* gene, leading to neuropathy and neuromuscular symptoms [117]. Several types of Schindler disease exist, and infants with the severe form often die before 4 years of age.

Many autosomal recessive mutations of the arylsulfatase A gene are known [119]. A common mutation leads to metachromatic leukodystrophy with a lack of oligomerization and instability of the sulfatase protein. Sulfatide accumulation in the central and peripheral nervous system and destruction of the myelin sheath occurs when lysosomal arylsulfatase A activity is less than 10% of the normal. The accumulation of sulfatide also affects other organs such as the kidney and gallbladder [120]. In the most severe forms, children suffer from dementia, muscle rigidity or paralysis, but many of the patients can live for several decades.

In sialidosis, gangliosides accumulate due to sialidase deficiency. Several different types of sialidosis occur. Mutations in the *NEU1* gene can cause neuraminidase deficiency associated with ocular abnormalities, bone pathologies, ataxia, mental decline and infantile death [121]. Another type of sialidosis is due to the absence of cathepsin A (PPCA) that prevents enzyme complex formation and thus depletes lysosomal activity of both sialidase NEU1 and β-galactosidase, causing accumulation of GM1 and other GSLs. Similarly, β-galactosidase deficiency also causes secondary neuraminidase deficiency [106].

Niemann–Pick disease and Farber disease are characterized by deficiency in acid sphingomyelinase and acid ceramidase, respectively. Thus, sphingomyelin accumulates in the nervous system of Niemann–Pick patients, together with hepatosplenomegaly and other organs. 

Secondary accumulation of smaller GSLs in lysosomes and endosomes in Niemann–Pick disease and other LSDs is seen. Although the mechanisms are still unknown, the shift in membrane lipids such as sphingomyelin may be responsible and contribute to the pathology. Deficiency of acid ceramidase is observed in Farber disease [95] with systemic lysosomal ceramide accumulation, spinal muscular atrophy and progressive epilepsy. Farber disease is extremely rare and occurs as several subtypes, depending on the level of residual enzyme activity, typically with deformed joints, subcutaneous nodules and progressive hoarseness [122]. It can occur early or later in life with accumulating ceramides and granulomas. Children with Farber disease may be born with hepatosplenomegaly and die before 2 years of age.

## 6. Pathophysiology of Lysosomal Storage Diseases

The accumulation of insoluble GSLs in the lysosomes affects membrane structures and function, and is responsible for initiating LSD pathology (Figure 2). The pathophysiology of LSDs primarily involves impairment of the degradation systems, i.e., autophagic-lysosomal system (ALS) and ubiquitin-proteasome system (UPS), which has a high degree of pathological similarity to neurodegenerative proteinopathies [123]. Autophagy is a complex cellular waste disposal system and includes macrophagy, microphagy and chaperone-mediated autophagy. Damaged organelles and dysfunctional proteins are engulfed by small vesicles known as autophagosomes that travel through the cytosol to a lysosome where the two organelles fuse for subsequent degradation. The process of autophagy is central for neurons and perturbation of this system is thought to contribute to neurodegeneration [124,125]. Additionally, the UPS is a major pathway for the clearance of misfolded and damaged proteins in the nucleus and cytoplasm, and involves ubiquitylation followed by proteasomal degradation by a multi-protein complex [123]. Considering the significant reduction in cellular waste disposal in LSDs, we propose the overwhelming lipid load may cause disruption of the lysosomal membrane rendering it unable to fuse with autophagosomes, which may indirectly lead to cell death.

### 6.1. Consequences of Intracellular Aggregation: Lysosomal Membrane Permeability

For most LSDs, the integrity of the lysosomal membrane is compromised by the storage material load due to dysfunctional lysosomal hydrolases. The lysosomal membrane becomes mechanically damaged as a result and causes lysosomal membrane permeability (LMP) [126,127,128]. Since the lysosome is home to many catabolic proteins, LMP may cause the release of proteases into the cytosol, which can initiate a complex pathway resulting in mitochondrial permeabilization. Cytochrome C may then leak from the mitochondria and activate caspases, which are powerful inducers of apoptosis. As well, accumulating gangliosides propagate ER stress responses and apoptotic signals causing further reinforcement of cellular pathology [129].

### 6.2. Inclusion Body Formation

Autophagy and the UPS are two lines of defense that prevent waste accumulation in the cell. A third possible defense mechanism that occurs frequently in LSDs and neurodegenerative diseases is inclusion body formation [130,131,132]. Accumulating lipids and misfolded proteins such as lysosomal enzymes together with ubiquitin can be found in inclusion bodies of neuronal cells of LSD, Alzheimer’s (AD) and Parkinson’s disease (PD) patients. These inclusion bodies may be harmful to the cell, and can inhibit clathrin-mediated endocytosis and the internalization of cell surface receptors [133]. Aggregates deposit in the cytoplasm or nucleus of the cell, and typically consist of ubiquitinated misfolded proteins. Neurons are particularly prone to inclusion body formation, as seen in AD and PD, perhaps because of their lack of cell division and thus lack of opportunity to dilute protein waste. In the case of SNCA, the precipitation of insoluble waste aggregates in inclusion bodies may reduce the concentration of toxic oligomers. However, inclusion bodies are also harmful to the cell [134]. Cytoplasmic aggregates of tau are capable of blocking the 20S and 19S proteasome particles of the UPS and prevent proper degradation of proteins [130]. With progressive accumulation of cytoplasmic inclusion bodies, cellular functions are disrupted which may indirectly cause cell death.

### 6.3. Cell–Cell Transmission of Pathology

One method of transmission of pathology between cells may be exocytosis of aggregated proteins/lipids into the extracellular space, which may then be endocytosed by other cells, initiating pathogenesis in neighboring cells. Lipid rafts can be taken up by neurons via endocytosis, and this can be upregulated in disease states [23]. Cells activate a number of secretory pathways that remove aggregated material such as Tau and Amyloid β (Aβ). This has been reported in cells with impaired lysosomal function. For example, *Neu1* knockout mice have increased exocytosis of Aβ in response to aggregated intracellular Aβ [135]. Another aggregated protein, SNCA, was shown to be internalized into human neuroglioma (H4) cells from the extracellular space by dynamin-dependent, clathrin-mediated endocytosis [136]. As a result of the accumulation of aggregated SNCA in neighboring cells, their lysosomal activity was significantly reduced and lysosomal morphology and function was changed. The high concentration of aggregated SNCA was able to seed aggregation of endogenous SNCA.

Receptor–ligand binding may also mediate cell–cell propagation of pathology. The laminin receptor (LRP/LR) has been found to facilitate the internalization of Aβ in AD [137]. Blockage of the receptor by antibodies significantly enhanced cell viability and proliferative ability. Recently, it has been found that the laminin receptor LRP1 is also capable of internalizing the Tau protein in AD and allows the spread of Tau between cells. The Tau protein is rich in Lys residues that bind to negatively charged residues of LRP1 [138].

### 6.4. Disruption of Autophagy

In a healthy cell, autophagy is a key mechanism for the clearance of cellular waste. Specifically, macrophagy involves engulfment of damaged organelles or dysfunctional proteins by membrane vesicles known as autophagosomes, which then fuse with the lysosome where hydrolases are exposed. However, it has been shown that these processes are impaired in LSDs [139]. In a Niemann–Pick mouse model, Purkinje cells exhibit clusters of aggregated ER, mitochondria and vacuoles within the cytoplasm suggesting a reduction in cellular waste clearance [140,141]. Autophagy disruption has also been observed in GD [142] and GM1 gangliosidosis [143] where neuronal processes and cell bodies were packed with inclusion bodies and large lysosomal volumes, as well as increased numbers of autophagic vacuole components. Therefore, the ALS may serve as a novel and important drug target for LSD treatment. In fact, amodiaquine and thiethylperazine are two novel drugs, which have been developed to restore autophagy in GM1 gangliosidosis by upregulating lysosomal hydrolases and activating the ALS in neuronal stem cells [143].

### 6.5. Alzheimer’s Disease

AD has been associated with altered ganglioside expression, although the mechanisms by which this occurs remain to be assessed. While total GSL levels are normal in AD brains, significantly higher levels of gangliosides with five or six sialic acid residues are present. As well, reduced levels of GM1 and GD1a (Figure 1E) were found in cholinergic neurons of post-mortem AD brains [144,145]. The pathophysiology of AD begins with abnormal folding and cleavage of amyloid precursor protein (APP) by β- and γ-secretases into small peptides that have a high propensity to aggregate and form senile plaques (Aβ) on the outside of the cell. Oligomeric Aβ peptides aggregate around lipid rafts embedded in neuronal cell membranes. Recently, it has been shown that GM1 interacts with Aβ peptides, which induces self-assembly of β-sheets leading to polymerization, formation of fibrils and ultimately neurodegeneration [146]. This may be problematic as polymerization of Aβ fibrils may disrupt extracellular trafficking and communication between cells. In contrast, GM1 oligosaccharides appear to prevent the aggregation of SNCA [79,82,147,148].

Aβ internalization is highly dependent on GSL and cholesterol concentration [149]. Saavedra et al. illustrated that sympathetic neurons internalize Aβ oligomers, which were colocalized with cholera toxin subunit B (CTxB), a marker for GM1 embedded in lipid rafts [150]. This suggests that Aβ is internalized via GM1 binding at lipid raft sites, and this triggers aggregation of Aβ plaques inside the cell. The clearance of intracellular plaques is drastically reduced in AD, likely due to failure of the ALS to manage such a high load of aggregated protein. Aβ accumulates, together with Aβ-ganglioside complexes and other aggregated proteins, in the endosomal/lysosomal compartments inside neurons. In AD patients, Aβ may be exocytosed to the extracellular matrix to begin propagation of extracellular pathology.

### 6.6. Parkinson’s Disease

In LSDs as well as in PD, SNCA can change from its native unfolded conformation to β-sheet-rich toxic, oligomeric and aggregated forms. The degradation of misfolded SNCA relies on the ALS and UPS. However, the presence of SNCA aggregates in neurons can impair autophagy, disrupt the UPS and activate microglia [151]. Due to an inability to degrade SNCA aggregates, fibrils and oligomers are sequestered into lipid vesicles named Lewy bodies. The pathological consequences for dopaminergic neurons in the substantia nigra are Fas-mediated cell death, mitochondrial dysfunction, oxidative stress, neuroinflammation and proteasome impairment [152].

Alterations in the concentration of neuronal gangliosides may trigger self-assembly of oligomeric SNCA aggregates [153]. Psychosines (Gal-sphingosine, lyso Gal-Cer) have been reported to facilitate SNCA fibrillation [154]. Another mechanism leading to SNCA aggregation is low activity of GCase (GBA) in lysosomes, causing accumulation of GlcCer, mimicking the symptoms of GD. Although only a small fraction of PD patients have mutations in the *UGCG* gene and GCase deficiency, they are the highest genetic risk factors for PD [101]. Several mutations to *LRKK2, VPS35* or *PARKIN* genes in PD have been associated with lysosomal stress, accumulation of abnormal autophagosomes and defective retrograde transport of endosomes [155]. There is also a link between the kinase activity of LRRK2 and GCase activity in neurons from PD patients [156].

Restoration of the lysosomal function and activation of GCase could be achieved in neuronal stem cells from PD patients using a small modulator compound [157]. Potential therapies could increase GCase activity, perhaps by acting as molecular chaperones. SNCA also binds to phospholipids and fatty acids, (e.g., phosphatidic acid and docosahexaenoic acid) and if present in high concentrations, these can induce SNCA aggregation [158].

SNCA has a central lipid-binding domain capable of binding GM1. N-acetylation of terminal Met1 appears to be essential for the SNCA–GM1 interaction [159,160]. It has been illustrated that GM1 reduces the fluidity of the plasma membrane, which may be the mechanism by which GM1 prevents the polymerization and fibrillization of SNCA [161].

Analysis of the substantia nigra of PD patients revealed that GM1, GD1a, GD1b and GT1b gangliosides were significantly reduced in neuromelanin-containing neurons [162]. This could be explained by reduced expression of the gene encoding B3GALT4, which normally adds Gal to GalNAc in the synthesis of GM1 or GD1b (Figure 1E). Significant improvement has been found in PD, but not AD patients, after treatment with intravenous GM1, and no serious adverse effects or anti-GM1 antibodies were noted after 24 weeks of treatment [163,164]. Clearly, endocytosis of SNCA is heavily dependent on GM1 binding to SNCA near the lipid rafts. The behavior of SNCA therefore seems to depend on the structure of associated membranes [151,165,166].

## 7. Therapeutic Approaches for Lysosomal Storage Diseases

Timely treatment of LSDs ideally would begin through newborn screening (NBS) programs [167]. Currently NBS is used for a small number of conditions. At present, the mainstay of treatment for most LSDs is either through enzyme replacement therapy (ERT) or substrate reduction therapy.

ERTs for many LSDs have been developed for mice and humans [10,168,169]. The enzyme must be targeted to the lysosome, which is typically accomplished with a Man-6-P tag, and usually does not cross the BBB. Protein degradation and immune response is a major barrier to the amount of available enzyme making regular treatments necessary. Successful examples of ERT include that for Fabry and GD [170].

Enzyme activation therapy is another potential treatment, although it is conditional on residual enzyme activity. Chaperones such as HSP70 may improve hydrolase stability and activity. There are also pharmacological chaperones designed for HEXA that stabilize protein conformation. This allows the enzyme to circumvent ER-associated protein degradation, and to travel to the lysosome where the acidic pH dissociates the complex yielding an active enzyme. The pyrimidine analog pyrimethamine is a molecular chaperone for remnant β-hexosaminidase activity in SD patient-derived fibroblasts [171]. Pyrimethamine was shown to bind to the active site of HEXB, and thus act as a substrate analog that stabilizes activity [172]. Protein modeling suggests that the analog can also bind HEXA.

Since GlcCer is the intermediate in the synthesis of all gangliosides, the inhibition of GlcCer synthase UGCG would prevent the accumulation and thus the natural functions of gangliosides. A number of inhibitors have been developed for substrate reduction therapy for LSDs [173]. N-butyldeoxynojirimycin inhibits UGCG and prevents GM2 accumulation in SD mice. The inhibitor was also successful for patients with mild GD, but not in severe infantile TSD. The hydrophobic and cationic UDP-Glc analog, Eliglustat, specifically inhibits UGCG by binding to the donor substrate binding site and has shown safe yet effective results for GD patients [174].

To overcome the limitations of current treatments, gene therapies have been in development as ‘promising’ and ‘one-time treatment’ potential therapeutics. Presently gene therapy aims to replace the mutated or missing gene with a functional gene to produce active enzyme in patient cells. In the future, gene editing using CRISPR-Cas9 or other technologies also hold promise.

Gene therapy methods can be divided into ex vivo and in vivo therapies. Ex vivo gene therapy involves extracting cells from the patient, introducing the correct gene in those cells and then administering them back to the patient. Within the past 20 years, a few ex vivo therapies have been investigated for LSDs. In 2004, Biffi et al. carried out a proof-of-concept study in mice, in which ex vivo transduction of hematopoietic stem cells with a lentiviral vector carrying lysosomal arylsulfatase A gene was capable of repopulating CNS microglia and peripheral endoneural macrophages. This treatment restored motor conduction, learning and coordination in a murine model of metachromatic leukodystrophy [175]. These results were later confirmed by the same group and the ex vivo therapy was brought to phase I/II clinical trials [176,177]. Three patients with metachromatic leukodystrophy were administered the transduced hematopoietic stem cells and exhibited stable engraftment, high expression of functional arylsulfatase A and arrested disease progression. The trial showed no indication of lentiviral vector integration near proto-oncogenes suggesting long-term therapeutic safety for future patients. A similar strategy is now being investigated for Krabbe disease, Fabry disease and many other disorders [178,179,180,181].

In vivo gene therapy has now taken the spotlight. Typically, in vivo gene therapy utilizes a virus for packaging and delivery of the corrected gene to the patient. In a preclinical investigation, adeno-associated viral (AAV) vector was used to deliver murine β-galactosidase (mβgal) to mβgal-deficient mice (GM1 gangliosidosis model), which significantly extended the survival of mice. This was attributed to reduction in lysosomal storage of GM1 in the thalamus, brainstem and spinal cord, as well as a marked decrease in astrogliosis throughout the CNS [182]. This year, AAV-human β-galactosidase therapy for treatment of GM1 gangliosidosis entered phase I/IIa human clinical trials and results are anticipated by the end of 2025. 

In vivo gene therapy for GM2 gangliosidosis has been quite proliferative in recent years. The first reports of viral vector delivery of *HEXA* was in 1996, where Akli et al. [183] found that transduction of TSD patient skin-derived fibroblasts with an adenoviral vector produced 40–84% of normal HEXA enzyme activity. Later, in vivo delivery of single transgenes *HEXA* or *HEXB* with AAV technology were found to successfully treat TSD and SD respectively [114,184]. However, administration of only one of the two genes (i.e., *HEXA* and *HEXB*) may limit heterodimer formation due to the availability of proportionate amounts of subunits. Therefore, two novel vectors have been designed as a solution: an artificial hybrid HEX enzyme called *HEXM* and the other, a bicistronic vector carrying both *HEXA* and *HEXB* in the same cassette [168,185,186]. In vivo testing of these vectors has shown marked reduction of GM2 ganglioside storage and a significant increase in survival of SD mice. Similarly, gene therapy is being developed for GSL-related LSDs like GM3 synthase deficiency, GD, Fabry and several other conditions.

## 8. Conclusions

We are just beginning to understand the pathophysiology of LSDs and related neurodegenerative diseases. One commonality is now quite apparent: pathology is heavily dependent on the cell’s membranes and their lipid components. It is necessary for the cell to maintain homeostasis of the large variety of GSLs, especially gangliosides in neuronal membranes, by tightly regulating GSL biosynthesis and degradation in the lysosomes. Membrane lipids interact with and control the functions of membrane proteins in cell proliferation, cell death, cell–cell interactions and neuronal activities. Therefore, perturbation in GSL balance causes severe pathological changes within the cell that may culminate in its demise. With virtually no viable treatments or targets to stop the disease progression of LSDs, patients and their healthcare team are desperate for translatable therapeutic options. Gene therapy for restoration of a functional enzyme is one option that shows great promise, especially in combination with other treatments that are based on the knowledge of GSL function and metabolism. Future research should investigate the regulation of GSL turnover and define possible targets in pathological conditions for identification of new therapies. If we can mobilize catabolic mechanisms to degrade aggregated and accumulated proteins, we could slow the progression of neurodegeneration, prolong patient survival and improve their well-being.

## Figures and Tables

**Figure 1 ijms-21-06881-f001:**
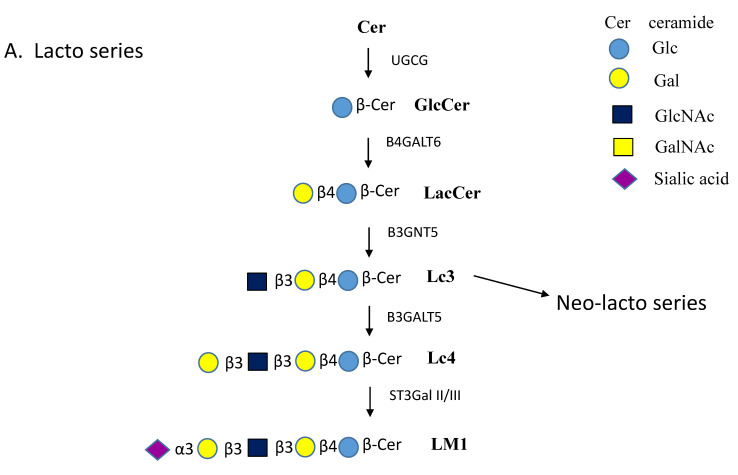
Pathways of glycosphingolipid (GSL) biosynthesis. (**A**). Lacto series GSLs. Lactosylceramide (LacCer) is the common intermediate for lacto, globo and ganglio series GSLs. The lacto series GSLs can have numerous different extensions and can carry glycan epitopes and antigens. Addition of GlcNAc to LacCer produces the precursor to all neo-lacto series GSLs, known as lactotriaosylceramide (Lc3). Lc3 is extended by the substitution of GlcNAc by a Galβ1-4 residue. The names of enzymes involved are indicated near the arrows. (**B**). Gal-ceramide series GSLs. Gal is added to ceramide by Cer-β-Gal-transferase UGT8 and reaction product GalCer can then undergo at least three different substitutions. (**C**). Globo series GSLs. The globosides are first produced by substitution of the Gal residue of LacCer with a Galα1-4 residue to yield globotriaosylceramide (Gb3). Addition of GalNAc in β1-3 linkage to Gb3 forms globotetraosylceramide (Gb4). The iso-globo series contains Galα1-3 instead of Galα1-4. (**D**). Ganglioseries-α GSLs. LacCer is the precursor for neutral GSLs, GA2 and GA1, and for all gangliosides. The addition of sialic acid in α2-6 linkage to GalNAc forms the α series of compounds. Here we show the synthesis of GD1α from GM1b. (**E**). Ganglio series a, b and c GSLs. The GM3 synthase ST3GAL V adds a sialic acid in α2-3 linkage to the Gal residue of LacCer to form GM3. The series that carries one sialic acid on Galβ1-4 is named ganglio a-series. Gangliosides having sialylα2-8sialylα2-3 substitutions are name ganglio b-series. The c-series have sialylα2-8sialylα2-8sialylα2-3 substitutions.

**Figure 2 ijms-21-06881-f002:**
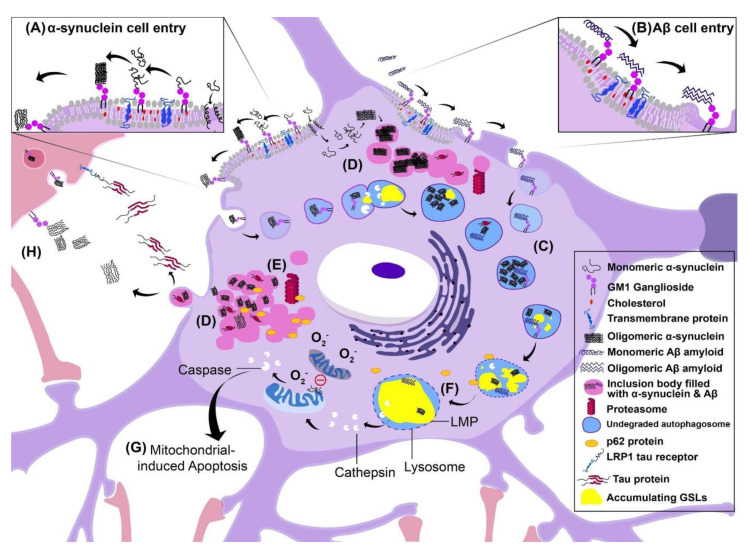
Schematic model of proposed pathophysiological mechanisms of lysosomal storage disorders (LSDs). The model is of the most affected cell type of LSDs, the neuron, and illustrates the events, which may indirectly or directly lead to cell death. A legend is presented on the right, which defines the symbols used in the figure. Several events are occurring simultaneously inside the diseased neuron: (**A**) α-synuclein cell entry, (**B**) Aβ cell entry, (**C**) accumulation of undegraded autophagosomes, (**D**) accumulation of cytoplasmic inclusion bodies filled with α-synuclein, Aβ or the Tau protein, (**E**) inhibition of the proteasome, (**F**) lysosomal membrane permeability changes, (**G**) mitochondrial-induced apoptosis (**H**) and cell–cell spreading of pathology. (**A**) α-synuclein enters the cell via GM1-receptor mediated endocytosis or by creating a pore in the membrane leading to the formation of inclusion bodies (seen in pink) that inhibit the proteasome and reduce degradation of materials in the cell. (**B**) Aβ amyloid enters the cell via the same method: GM1-mediated endocytosis. (**C**) Autophagosomes fill with undegraded material such as the accumulating proteins α-synuclein, Aβ and Tau. (**F**) Due to defective lysosomal hydrolases, GSLs accumulate in the lysosome and promote lysosomal membrane permeability (LMP). (**G**) Eventually the lysosome expels its contents including cathepsins. This induces mitochondrial release of caspases, which in turn cause apoptosis. Additionally, monomeric α-synuclein is capable of inhibiting mitochondrial fusion. As a result, the dying mitochondria degrade in the cytoplasm, releasing copious amounts of reactive oxygen species inducing cell death. (**H**) As the cell fills with undegraded autophagosomes, and harmful inclusion bodies, the cell activates secretory pathways that lead to the exocytosis of intracellular material to reduce the load, followed by endocytosis in adjacent cells. This provides a route for pathology to spread to adjacent cells. Harmful aggregates such as from Tau, α-synuclein and Aβ can be endocytosed into the adjacent cell. Specifically, Tau can interact with the receptor LRP1 to allow cell entry.

**Table 1 ijms-21-06881-t001:** Human transferases involved in the biosynthesis of glycosphingolipids.

Enzyme Names	Uniprot No.	CAZy Family
Cer β-Glc-transferase, GlcCer synthase, UGCG	Q16739	GT21
Cer β-Gal-transferase, GalCer synthase CGT, UGT8	Q16880	GT1
Cerebroside sulfotransferase, Gal3ST1, CST	Q99999	-
CerGlc β1,4Gal-transferase, B4GALT6, LacCer synthase	Q9UBX8	GT7
GalNAc β1,3Gal-transferase, B3GALT4, GM2 synthase	O96024	GT31
GlcNAc β1,3Gal-transferase, B3GALT5	Q9Y2C3	GT31
Gal α1,4Gal-transferase, A4GALT, Gb3 synthase	Q9NPC4	GT32
LacCer β1,4GalNAc-transferase, B4GALNT1, GD2 synthase	Q00973	GT12
Gal β1,3GalNAc-transferase, B3GALNT1, Gb4 synthase	O75752	GT31
Gal β1,3GlcNAc-transferase, B3GNT5	Q9BYG0	GT31
Gal α2,3Sialyltransferase, ST3Gal I	Q11201	GT29
ST3Gal II	Q16842	GT29
ST3Gal III	Q11203	GT29
ST3Gal IV	Q11206	GT29
GM3 synthase, ST3Gal V	Q9UNP4	GT29
ST3Gal VI	Q9Y274	GT29
GalNAc α2,6sialyltransferase, ST6GALNAC V	Q9BVH7	GT29
Sialyl α2,8sialyltransferase, ST8Sia I	Q92185	GT29
ST8Sia III	O43173	GT29
ST8Sia V	O15466	GT29

**Table 2 ijms-21-06881-t002:** Disorders of glycosphingolipid biosynthesis.

Name of Disease	Gene	Enzyme	Affected GSL	MIM Phenotype No.
Spastic Paraplegia (Type 26)	*B4GALNT1*	β1,4GalNAc-transferase	GM2, GM1, etc.	609195
West Syndrome	*ST3GAL III*	α2,3Sialyltransferase 3	GM1, GD1a, etc.	615006, 611090
Amish Infantile Epilepsy	*ST3GAL V*	α2,3Sialyltransferase 9 (GM3 synthase)	LacCer	609056

**Table 3 ijms-21-06881-t003:** Human hydrolases acting on glycosphingolipids and ceramide.

Enzyme Names	Uniprot No.	CAZy GH/clan	Activator
Glucoceramidase, GCase, GBA1	P04062	GH30 / GH-A	SapC
β-Galactoceramidase, GALC, Gal-Cer β-galactosidase	P54803	GH59 / GH-A	SapA
Arylsulfatase A, ARSA	P15289	-	SapB
α-Galactosidase A, GLA	P06280	GH27 / GH-D	SapB
β-Galactosidase, GLB1, BGAL	P16278	GH35 / GH-A	SapB
β-Hexosaminidase HEXA	P06865	GH20 / GH-K	GM2-AP
β-Hexosaminidase HEXB	P07686	GH20 / GH-K	GM2-AP
α-N-Acetylgalactosaminidase NAGA	P17050	GH27 / GH-D	-
Sialidase NEU1	Q99519	GH33 / GH-E	-
Sialidase NEU2	Q9Y3R4	GH33 / GH-E	-
Sialidase NEU3	Q9UQ49	GH33 / GH-E	-
Sialidase NEU4	Q8WWR8	GH33 / GH-E	-
Acid ceramidase, ASAH1	Q13510	-	SapD

**Table 4 ijms-21-06881-t004:** Disorders of glycosphingolipid degradation.

Name of Disease	Gene	Enzyme	Affected GSL	MIM Phenotype No.
GM1 Gangliosidosis	*BGAL*	β-galactosidase	GM1	Type I: 230500 Type II: 230600 Type III: 230650
GM2 Gangliosidosis Variant B: Tay-Sachs Disease	*HEXA*	β-hexosaminidase A/S (HexA/S) (α subunit)	GM2	272800
GM2 Gangliosidosis Variant 0: Sandhoff Disease	*HEXB*	β-hexosaminidase B (HexB) (β subunit)	GM2	268800
GM2 Gangliosidosis Variant AB: GM2-AP Deficiency	*GM2A*	GM2-activator protein (GM2-AP)	GM2	272750
Fabry Disease	*GLA*	α-galactosidase A	Globosides, Blood group B	301500
Gaucher Disease	*GBA1*	β-glucoceramidase I (GCase I)	GlcCer	Type I: 230800 Type II: 230900 Type III: 231000
Krabbe Disease	*GALC*	β-galactoceramidase SapA	GalCer	245200 606463
Metachromatic Leukodystrophy	*ARSA*	Arylsulfatase A	Sulfatide	250100
Saposin deficiency	*PSAP*	Prosaposin (Saposin precursor protein)	GSLs	611721
Galactosialidosis, PPCA Deficiency	*CTSA*	β-galactosidase Sialidase I (NEU1) Cathepsin A	GM1	256540
Sialidosis	*NEU1*	Sialidase I (NEU1)	Sialylated GSLs	256550
Niemann-Pick Disease	*SMPD1*	Acid sphingomyelinase	Sphingosine	Type A: 257200 Type B: 607616
Schindler Disease	*NAGA*	α-N-acetylgalactosaminidase B (α-NAGAL)	Lac-Cer, Blood group A	609241
Farber Disease	*ASAH1*	Acid ceramidase	Ceramides	228000

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
