# Peer review of "Metabolism of Glycosphingolipids and Their Role in the Pathophysiology of Lysosomal Storage Disorders"

_ijms, 2020, doi:10.3390/ijms21186881_

Round 1

Reviewer 1 Report

The topic of the article is to wide for a review article to be published in IJMS.  At present, it looks like a book chapter. The various sections and the related references are not balanced nor homogeneous, and several misunderstandings occur. I suggest the authors to focus on some aspects, which should be treated in detail, deleting many other sections that are not properly addressed. I also suggest to pay more attention to each reference, and to make the text-reference relationship more logical. At present, it is not clear why the reference of a sentence is a review article or an original article, and why a twenty years old reference is used instead of a recent one. Many relevant sentences and concepts are reported without any reference, and this is not acceptable. As a general rule, I would suggest that the reference list should contain a vast majority of articles published within the last 10 years, while those older than 15 years should represent only relevant exceptions. In my opinion, the article should be rewritten focusing on selected topics, the sections re-organized, and the references updated and carefully selected.

The following are relevant examples, but almost all the text suffers the same problems.

Lines 49, 282-283, and Tables: why B4GALT6 and not B4GALT5, or both? There are relevant recent papers suggesting the predominant role of B4GALT5 as LacCer synthase in mice, please complete and correct.

Lines 90-91: Ref. 8 (1992) deals with the lipid composition of GM3, but the sentence deals with a-b-c series gangliosides

Lines 92-94: why a methodological article dated 1973 and a review article with very general topic dated 2011? Are they pertinent to the potential developmental pattern of gangliosides? In human or model systems? How do models match with the features of newborn carrying defects in ganglioside biosynthesis?

Lines 149-151: “It is widely known …” is a sentence usually not acceptable in a scientific article, particularly in controversial field as that of gangliosides and neurodevelopment. No reference cited versus a big question placed by patients lacking GM3 synthase and related ganglio-series gangliosides: their brain looks normal at delivery. This is a topic that must be considered in detail or instead deleted.

Lines 213 – 239: The localization and topology of biosynthetic enzymes is not clear: please detail step by step with proper references and potentially contradictory data.

Lines 313-338:  The section should be rewritten taking into account carefully the results presented by the literature, with particular attention to the many patients reported to suffer B4GALNT1 deficiency.  The results about post-mortem brain studies must be accompanied by the proper reference.

Lines 419-431: Description and references related to Gaucher disease are inadequate, particularly with respect to the following section dealing with gangliosidosis. This section must be completely rewritten in a proper and detailed manner or deleted making simple reference to recent exhaustive review articles.

Lines 505-513: The section dealing with Farber and Niemann-Pick diseases lacks any reference or relevant concept/description. It should be deleted or completely rewritten.

Section 6 lines 514-555: 8 out 12 references cited (90-101) are 10 years old or older.

Minor points

Abbreviations: please refer to the HUGO nomenclature for genes and proteins; avoid previous non-standard abbreviations for enzymes in the text, tables and figures.

For the diseases caused by congenital defects of glycosylation, use the standard name of the disease based on the gene name followed by -CDG (ST3GAL5-CDG, B4GALNT1-CDG etc.).

Line 71: use an abbreviation for Fucose as for the other sugars

Author Response

Reviewer 1:

Comments and Suggestions for Authors

The topic of the article is to wide for a review article to be published in IJMS.  At present, it looks like a book chapter. The various sections and the related references are not balanced nor homogeneous, and several misunderstandings occur. I suggest the authors to focus on some aspects, which should be treated in detail, deleting many other sections that are not properly addressed. I also suggest to pay more attention to each reference, and to make the text-reference relationship more logical. At present, it is not clear why the reference of a sentence is a review article or an original article, and why a twenty years old reference is used instead of a recent one. Many relevant sentences and concepts are reported without any reference, and this is not acceptable. As a general rule, I would suggest that the reference list should contain a vast majority of articles published within the last 10 years, while those older than 15 years should represent only relevant exceptions. In my opinion, the article should be rewritten focusing on selected topics, the sections re-organized, and the references updated and carefully selected.

Response: We very much appreciate the comments and have made an effort to rewrite the review accordingly. We expanded the content of sections to make them more complete. We paid attention to the text-reference relationship and rearranged some of the topics. We added new (last 10 years) additional references to back up the statements, and converted some of the older references to newer ones. Some older references are classical exceptional papers or the first discovery of concepts that are worth being acknowledged. We believe that the sections of this review are relevant to lysosomal storage diseases, their origins, pathology and treatments.

The following are relevant examples, but almost all the text suffers the same problems.

Lines 49, 282-283, and Tables: why B4GALT6 and not B4GALT5, or both? There are relevant recent papers suggesting the predominant role of B4GALT5 as LacCer synthase in mice, please complete and correct.

Response: We added the information on B4GALT5 in mice.

Lines 90-91: Ref. 8 (1992) deals with the lipid composition of GM3, but the sentence deals with a-b-c series gangliosides

Response: We replaced Ref 8 with newer references.

Lines 92-94: why a methodological article dated 1973 and a review article with very general topic dated 2011? Are they pertinent to the potential developmental pattern of gangliosides? In human or model systems? How do models match with the features of newborn carrying defects in ganglioside biosynthesis?

Response: We discussed the similarity between human disease and mouse models in section 2.2 and in sections 3.3 and 5.3 and added newer references. The Tettamanti paper 1973 is a classical paper and the methods are still being followed now.

Lines 149-151: “It is widely known …” is a sentence usually not acceptable in a scientific article, particularly in controversial field as that of gangliosides and neurodevelopment. No reference cited versus a big question placed by patients lacking GM3 synthase and related ganglio-series gangliosides: their brain looks normal at delivery. This is a topic that must be considered in detail or instead deleted.

Response: The English has been edited throughout. We added a sentence on the GM3 synthase deficiency in mice in comparison to humans in section 3.3.

Lines 213 – 239: The localization and topology of biosynthetic enzymes is not clear: please detail step by step with proper references and potentially contradictory data.

Response: We discussed the localization of the enzymes of the individual steps of biosynthesis in sections 3, 3.2 and 4.

Lines 313-338:  The section should be rewritten taking into account carefully the results presented by the literature, with particular attention to the many patients reported to suffer B4GALNT1 deficiency.  The results about post-mortem brain studies must be accompanied by the proper reference.

Response: We added a paragraph on B4GALNT deficiency in section 3.3 and added more references.

Lines 419-431: Description and references related to Gaucher disease are inadequate, particularly with respect to the following section dealing with gangliosidosis. This section must be completely rewritten in a proper and detailed manner or deleted making simple reference to recent exhaustive review articles.

Response: We expanded the discussion on Gaucher disease and updated references. We also added therapy trials on Krabbe in section  5.1 and GM1 gangliosidosis in section 5.2 with new references.

Lines 505-513: The section dealing with Farber and Niemann-Pick diseases lacks any reference or relevant concept/description. It should be deleted or completely rewritten.

Response: We agree. The discussion on Niemann Pick and Farber was rewritten and several references were added.

Section 6 lines 514-555: 8 out 12 references cited (90-101) are 10 years old or older.

 Response: The older references were replaced by newer ones.

Minor points

Abbreviations: please refer to the HUGO nomenclature for genes and proteins; avoid previous non-standard abbreviations for enzymes in the text, tables and figures.

Response: We changed the names of enzymes in Text, Tables and Figures to include the HUGO nomenclature. We agree that it is confusing to have different names for the same protein and that a standard name should prevail. Unfortunately, cited papers have enzyme names in the title that are not from HUGO.

For the diseases caused by congenital defects of glycosylation, use the standard name of the disease based on the gene name followed by -CDG (ST3GAL5-CDG, B4GALNT1-CDG etc.).

Response: We included the -CDG names.

Line 71: use an abbreviation for Fucose as for the other sugars

Response: It is now done.

Reviewer 2 Report

In this article, Ryckman et al provide an in depth review of glycosphingolipid (GSL) structure, function, metabolism, microbial interactions, and discuss genetic defects of GSL biosynthesis. They provide detailed information on the pathophysiology of disease as well as therapeutic approaches for lysosomal storage disorders. The manuscript is very well written with no obvious grammatical or typographical errors and is of scientific interest to the community. I have only minor suggestions for the authors.

  1. Recommend to add the MIM phenotype number after the names of the genetic disorders upon first introduction (or within Table 2 and Table 4).
  2. Recommend to change the words ‘Mutated Gene’ and ‘Defective Enzyme’ in Tables 2 and 4 to Gene and Enzyme, respectively
  3. Recommend to change the word mutations to pathogenic variants in section 5 when referring to disease-causing variants.
  4. Recommend to change the resolution/font size of Fig 1 B to match Fig 1A,C,D and E

Author Response

Reviewer 2:

Comments and Suggestions for Authors

In this article, Ryckman et al provide an in depth review of glycosphingolipid (GSL) structure, function, metabolism, microbial interactions, and discuss genetic defects of GSL biosynthesis. They provide detailed information on the pathophysiology of disease as well as therapeutic approaches for lysosomal storage disorders. The manuscript is very well written with no obvious grammatical or typographical errors and is of scientific interest to the community. I have only minor suggestions for the authors.

            Response: Thank you for reviewing this review.

  1. Recommend to add the MIM phenotype number after the names of the genetic disorders upon first introduction (or within Table 2 and Table 4).

Response: The MIM phenotype numbers were added in Tables 2 and 4

  1. Recommend to change the words ‘Mutated Gene’ and ‘Defective Enzyme’ in Tables 2 and 4 to Gene and Enzyme, respectively

Response: Tables 2 and 4 were corrected.

  1. Recommend to change the word mutations to pathogenic variants in section 5 when referring to disease-causing variants.

Response: It is now done.

  1. Recommend to change the resolution/font size of Fig 1 B to match Fig 1A,C,D and E

Response: The figures have the same resolution now in the text. However, figures (in powerpoint) are submitted separately.

Round 2

Reviewer 1 Report

The revised version is very similar to the original one. No relevant modifications in the approach, nor in the text or figures. There are only insertions of text and references that do not respond to the criticisms. My judment is thus similar to the previous one.